# Interventions Preventing Vaginitis, Vaginal Atrophy after Brachytherapy or Radiotherapy Due to Malignant Tumors of the Female Reproductive Organs—A Systematic Review

**DOI:** 10.3390/ijerph18083932

**Published:** 2021-04-08

**Authors:** Adrianna Wierzbicka, Dorota Mańkowska-Wierzbicka, Stanisław Cieślewicz, Marta Stelmach-Mardas, Marcin Mardas

**Affiliations:** 1Department of Obesity Treatment, Metabolic Disorders and Clinical Dietetics, Poznan University of Medical Science, Szamarzewskiego 84 St, 60-569 Poznan, Poland; duniak1@op.pl; 2Department of Gastroenterology, Dietetics and Internal Medicine, Poznan University of Medical Sciences, Heliodor Święcicki Hospital, Przybyszewskiego 49, 60-355 Poznań, Poland; dmankowska.wierzbicka@gmail.com; 3Department of Oncology, Poznan University of Medical Science, Szamarzewskiego 82/84 St, 60-569 Poznan, Poland; stcieslewicz@gmail.com (S.C.); marcin.mardas@ump.edu.pl (M.M.)

**Keywords:** women’s health, reproductive health, vagina, vaginal atrophy, radiotherapy, brachytherapy, endometrial cancer, cervical cancer, women’s mental health

## Abstract

Background: Radiotherapy, as a method of treatment of cervical and uterine cancers, may induce severe late-onset vaginal side effects. Unfortunately, little evidence on the management of adverse effects has been presented. This study aimed to evaluate the available interventions which reduce symptoms of vaginitis and vaginal atrophy by improving dyspareunia, mucosal inflammation, vaginal pH and vaginal dryness in women who have undergone brachytherapy or radiotherapy due to uterine or cervical malignancies. Materials and methods: A comprehensive literature search was performed following PRISMA guidelines. The systematic search was conducted using electronic databases, namely Scopus, Web of Science and PubMed, between October and November 2020 to identify randomized controlled trials (RCT) and, prospective randomized studies (PRS). Results: The analyzed population consists of 376 patients with uterine or cervical cancer, treated with hyaluronic acid, vitamin A, vitamin E, alpha-tocopherol acetate and dienestrol. Intervention with HA along with vitamin A and vitamin E revealed advantage in endpoints such as reduced dyspareunia, vaginal mucosal inflammation, vaginal dryness, bleeding, fibrosis and cellular atypia. Administration of alpha-tocopherol acetate reduced vaginal mucosal inflammation and improved vaginal acanthosis, whereas dienestrol resulted in reduced dyspareunia, vaginal caliber and bleeding. Conclusions: Vaginal suppositories were found to be clinically effective at the management of late-onset vulvovaginal side effects after radiotherapy.

## 1. Introduction

Globally, up to 50–60% of women experience symptoms of vulvovaginal atrophy [1]. The clinical signs associated with these lesions are diverse and incorporate recurrent urinary tract infections, dryness, dyspareunia and irritation. According to the literature, vaginitis is defined as “any condition with symptoms of abnormal vaginal discharge, odor, irritation, itching, or burning” [2] and vaginal atrophy as “a condition in which the tissues lining the inside of the vagina (birth canal) become thin, dry, and inflamed” [3]. All these manifestations when they have a chronic progressive course may significantly impact women’s health and quality of life (QoL).

The diagnosis of gynecological cancer may have a tremendous effect on a woman’s sexuality. For the majority of people, the satisfaction derived from the sexual experience is considered a fundamental part of a healthy and fulfilled life. Sexual functioning involves the complex interactions among the vascular system, the nervous system, the endocrine system and psychological factors. Nevertheless, for many women, it is more than the ability to have sexual intercourse. It acts as an integral component of the female identity which manifests as the perception of personal body image, gender identity, sexual orientation, fertility and femininity. Moreover, it serves as the core of a woman’s personality and sexual self-concept which is expressed and experienced through behaviors, beliefs, values, fantasies and relationships [4,5]. Gynecological malignancies can be treated with surgeries, chemotherapy and radiation, either solely or in combination. The cytotoxic chemotherapy has been associated with a number of side effects including alopecia, fatigue and decreased libido which result in altered self-esteem and reduced sexual interest [6]. Brachytherapy is another technique suggested for patients with diverse types of cancer including endometrial and cervical cancer [7]. This procedure reduces the risk of habitual tissue complications, by focally providing high dose radiotherapy to the malignancies [7]. Unfortunately, radiation may impair vaginal mucosa and therefore can provoke the onset of vaginal stenosis and fibrosis. These outcomes are associated with severe sexual dysfunction, poor pregnancy outcomes, fear of sexual intercourse and even infertility [8]. Even though they may be cured, these women will suffer from the late-onset side-effects of cancer management that will impact their QoL. Among women with endometrial and cervical cancer, the following aftereffects appear to be the most commonly reported: sexual dysfunction, vaginal dryness, vaginal atrophy, peripheral neuropathy and cognitive changes [9]. As a consequence, these bodily responses might result in emotional deviations and provoke feelings of anxiety. Furthermore, the physiological responses related to the sexual response cycle including sexual desire, arousal and orgasm can be markedly affected [4,5]. Unfortunately, little attention has been paid to the resultant sexual complications that women experience. Thus, adequate management including counseling and education must be implemented to improve sexuality and sexual functioning. The globally accepted international standards recommend vaginal lubricants as a first-line treatment option [1]. They are not only convenient to use but also provide the relief from vaginal discomfort and dryness during sexual intercourse. Furthermore, they exert long-lasting effects in alleviating vulvovaginal symptoms by ameliorating moisturizing of the vaginal epithelium [10]. 

Therefore, the aim of this systematic review is to appraise and identify the effective methods that would provide a relief to vaginitis and vaginal atrophy in women who have undergone brachytherapy or radiotherapy due to uterine or cervical malignancies. Moreover, this review explores the influencing factors of vaginal suppositories on vulvovaginal lesions through the improvement of dyspareunia, mucosal inflammation, vaginal pH and vaginal dryness.

## 2. Materials and Methods

### 2.1. Search Strategy and Study Selection 

A comprehensive literature search was performed following the Preferred Reporting Items for Systematic Reviews and Meta-Analysis (PRISMA). The systematic search was conducted through the use of electronic databases: Scopus, Web of Science and PubMed between October and November 2020 to identify randomized controlled trials (RCT) and prospective randomized studies (PRS). 

Search terms included: (jojoba OR coconut oil OR aloe vera OR Aloe barbadensis leaf juice OR vitamin E OR tocopherol acetate OR Lactobacillus OR olive oil OR vaginal estrogen OR vaginal hyaluronic OR vaginal topical estriol OR Replens OR K-Y Liquibeads OR Astroglide OR Sliquid OR K-Y Jelly OR Hyalofemme OR Silk-E OR vaginal lubricant OR vaginal moisturizer OR Euterpe oleracea pulp powder OR flaxseed extract OR locust beam gum OR Estrace OR Premarin OR Vagifem OR Estring OR Femring OR Imvexxy OR camomile OR vitamin A OR vitamin E oil OR Lubafax OR LMWHA OR low molecular weight hyaluronic acid OR Conjugated equine estrogen cream OR CEE cream OR CE cream OR C.E.S OR ospemifene OR Osphena OR Prasterone OR Intrarosa) AND (vaginitis OR Atrophic vaginitis OR vaginal atrophy) AND (brachytherapy OR curietherapy OR internal radiation OR Implant Radiotherapy OR Interstitial Radiotherapy OR Intracavity Radiotherapy OR radiotherapy OR X-ray therapy OR radiation treatment OR radiation therapy OR targeted radiotherapy OR targeted radiation therapy) AND (endometrial cancer OR endometrial tumor OR endometrial malignancy OR endometrial neoplasia OR Cancer of Endometrium OR Cancer of the Endometrium OR Carcinoma of Endometrium OR Endometrial Carcinoma OR Endometrium Cancer OR endometrioid tumor OR endometrioid cancer OR endometrioid carcinoma OR endometrioid neoplasia OR endometrioid malignancy OR endometrial stromal tumor OR endometrial stromal cancer OR endometrial stromal neoplasia OR endometrial stromal malignancy OR endometrioid adenocarcinoma OR cervical malignancy OR cervical carcinoma OR uterine cervical cancer OR Cancer of Cervix OR Cancer of the Cervix OR Cancer of the Uterine Cervix OR Cervical Cancer OR Cervical Neoplasms OR Cervix Cancer OR Cervix Neoplasms).

The inclusion criteria were articles published in the English language carried out only in adult women (aged over 18) who underwent brachytherapy or radiotherapy due to malignant tumors of the female reproductive organs involving endometrium and cervix. The criteria for exclusion were as follows: articles available only in abstract form, where no possible contact with authors could have been made, conference publications, case series, editorials, case reports, letters, comments and studies performed in specific groups of patients (e.g., patients with ovarian tumors, subjects who have not received brachytherapy or radiotherapy, pediatric patients or males).

### 2.2. Data Extraction

Two authors independently applied the risk of bias tools and resolved differences by discussion. The identified studies were screened based on titles and abstracts, and full texts of potentially applicable studies were retrieved. Additionally, the references for distinguished articles were checked for relevance. The following data were extracted: first author’s name, year of publication, country, study design, control and intervention group size, mean age, type and stage of cancer, intervention dosage and duration of time.

### 2.3. Outcomes and Measures

As the main outcomes, we selected dyspareunia defined as genital pain that can be experienced before, during or after intercourse [11]; mucosal inflammation defined as an inflammation of the mucosa with burning or tingling sensation [12]; dryness defined as loss of vaginal secretions and lubrications of the vaginal wall [13]; bleeding defined as any vaginal bleeding unrelated to normal menstruation [14]; and vulvar fibrosis defined as growth of adhesions from fibrous tissue in the vagina [15].

### 2.4. Risk of Bias Tool 

To assess the methodological quality of the studies included in this review, we used the “Cochrane Handbook of Systematic Reviews of Interventions” [16].

### 2.5. Statistical Analysis

Selected studies were checked for statistical methods. We focused on sample size calculation, the set of *p*-values and statistical tests reported. Effect of intervention on vaginal parameters was expressed either as percentage changes in selected measures or as the differences in absolute values. *p*-value in all selected studies was set at the level of 0.05. 

To combine the results of the individual studies, we attempted to perform a meta-analysis using a random-effects model, which allowed for true effect variation between studies (Review Manager (RevMan) V5.4; the Nordic Cochrane Centre, the Cochrane Collaboration, Copenhagen, Denmark). However, due to the high heterogeneity of selected studies, the quantitative synthesis could not be done. 

## 3. Results

### 3.1. Study Selection and Characteristics of the Study Population

A flow chart showing the study selection is provided in Figure 1. In total, 40 citations were identified by the database search. After removing duplicates, the eligibility of titles and abstracts were assessed by two independent reviewers. The full texts of 11 articles were retrieved, and seven were further excluded based on the reasons presented in Figure 1. Finally, four studies [17,18,19,20] with a total of 376 participants were included in the qualitative synthesis; three of them were RCTs [17,18,19] and one was PRS [20] with a duration ranging up to 40 weeks [19], published during 1971–2018. The majority of these studies were conducted in Italy [17,18,20], except for one study which was performed in the USA [19]. The mean age of enrolled patients was 52.9 years old. All participants were diagnosed with either cervical [17,18,19,20] or endometrial cancer [18]. Malignancy staging was classified according to the International Federation of Gynecology and Obstetrics (FIGO). The following intervention methods were used: hyaluronic acid [17,20], vitamin A [17,20], vitamin E [17,20], alpha-tocopherol acetate [18] and dienestrol [19]. Detailed characteristics of the selected studies are shown in Table 1 and Table 2. Risk of bias are presented in Figure 2. Performance biases were high or unclear risk due to the fact that studies were not placebo controlled. Additionally, detection bias presented a 75% “unclear” risk. Detection bias is related to the blinding of the assessors to the study results, and, although the Cochrane manual states that their blinding does not ensure success, lack of blinding could bias the study results. In any case, the predominant classification of this type of bias, in particular, was “unclear risk” and not “high risk”, which is associated with a lack of information regarding this bias on the part of the authors, rather than a possible bias in the results.

### 3.2. Effect of Topical Interventions on Vaginal Changes and Clinical Outcomes

All of the studies showed the superiority of vaginal interventions products versus no treatment. The gynecological inspection was performed in all studies [17,18,19,20]. In four of the studies, the pain severity was assessed by visual analog scale (VAS) [17,18,20]. One study [19] evaluated the ranking of symptoms using the Adverse Event Reporting Questions. Moreover, one study [18] estimated the tissue toxicity caused by radiation using the Radiation Therapy Oncology Groups Scoring System. One study [18] utilized the histological criteria to assess microscopical changes. A vaginal biopsy was performed in two studies [18,20]. Intervention with hyaluronic acid along with vitamin A and vitamin E [17,20] revealed advantage in endpoints such as reduced dyspareunia, vaginal mucosal inflammation, vaginal dryness, bleeding, fibrosis and cellular atypia. Moreover, the administration of alpha-tocopherol acetate [18] reduced vaginal mucosal inflammation and improved vaginal acanthosis. In addition, dienestrol [19] resulted in reduced dyspareunia, vaginal caliber and bleeding. The detailed characteristics of the selected studies are shown in Table 3 and Table 4.

## 4. Discussion

To begin with, patients with newly diagnosed gynecological cancers need optimal care management. The treatment of these malignancies uses a multimodal approach comprised of surgical excision, chemotherapy and radiotherapy including brachytherapy [21]. Unfortunately, each of these forms of treatment might cause consecutive long-term side effects such as atrophic vaginitis as a result of radiation therapy [22]. Alternatively, numerous vitamins, in different combinations, including vitamins A [17,20] and E, alpha-tocopherol [23] and hyaluronic acid [24], have been used in the management of vaginal stenosis and vaginal atrophy and have displayed a beneficial effect.

Vaginal lubrication is a complex physiological process which involves the interaction between the macro- and microvascular anatomy of the female vagina, hormonal stimuli and nervous system response [25]. This homeostasis is crucial for the adequate maintenance of the vaginal barrier which includes the mucosal epithelial cells, structural proteins and vaginal microbiota. It should be noted that any alterations related to the vaginal health may have detrimental effects on the increased risk of the adherence and entry of potentially harmful bacteria and viruses, and thus associated infections [26]. In addition, it has been reported that loss of vaginal lubrication and vaginal elasticity results in painful sexual intercourse [27]. Furthermore, research interest has progressively increased towards the drastic changes in the vaginal health in endometrial and cervical cancer survivors treated with radiation therapy [28]. The study results show an increased fraction of high density collagen (fibrosis) and vaginal mucosal atrophy in cervical cancer survivors [28]. Moreover, the majority of patients reported painful sexual intercourse along with altered vaginal elasticity [29].

### 4.1. Characteristics of the Study 

The most frequently investigated vaginal suppository intervention was a combination of hyaluronic acid, vitamin A and vitamin E [17,20]. This amalgamation regulates all steps of the inflammatory cascade through the enhancement and acceleration of physiological mucosal regeneration. Delia et al. [17] reported a significant reduction in mucosal inflammation, vaginal dryness and dyspareunia. Moreover, an improvement in terms of vaginal bleeding, fibrosis and cellular deformity was observed [20]. The present findings concur with the results presented by Constantino et al. [30], who also revealed burning and itching after the same intervention. Additionally, McLaren et al. [31] reported that the long-term application of vitamin E can improve symptoms of vulvovaginal lesions and reduce dyspareunia by enhancing atrophic scars in the lower reproductive tract. The alleviation of vulvovaginal lesions might be associated with biological properties of tocopherols. They influence the number of gonadotrophic hormones (follicle-stimulating hormone, luteinizing hormone and testosterone) and, therefore, indirectly alter the levels of estrogen and progesterone [32]. These endocrine neurotransmitters have a critical effect on vaginal thickness as well as vaginal epithelium [33]. Another worthwhile study [34] investigated the effect of vitamins E and D administration on vaginal atrophy in patients with breast cancer who were receiving tamoxifen, showing a significantly beneficial effect interlinked with reduction in vaginal pH. In another study, postmenopausal patients with vaginal atrophy underwent the treatment with the vaginal suppository of vitamin E, hyaluronic acid, phytoestrogen from *Humulus lupulus* extract and liposome [35]. In the aforementioned study, significant improvements in dyspareunia, redness, inflammation, burning, edema and itching were observed [35]. It is worth mentioning that, after radiotherapy, the capacity to renew the vaginal epithelium significantly decreases, leading to increased mucosal fragility and vaginal pain [36]. The process of vaginal mucosal epithelial repair is complex and among others involves angiogenesis. Interestingly, it has also been shown that application of hyaluronic acid upregulates the expression of vascular endothelial growth factor (VEGF) as well as serine/threonine protein kinase B (P-AKT) [37]. Activation of this cascade leads to release of nitric oxide synthase and, hence, promotes angiogenesis. Relatedly, a study has shown synergy between VEGF and hyaluronic acid [38]. The topical application of this molecular compound exerts hydrating properties and, therefore, can attenuate dryness associated pruritis [39]. These results are in line with a study by Grimaldi et al. [40], where high molecular weight hyaluronic acid alleviated erythema, vaginal atrophy and vaginal dryness in post-menopausal patients. Another study indicated the effectiveness of hyaluronic acid vaginal tablets in the reduction of urogenital atrophy in breast cancer patients who have experienced post-chemotherapy or post-hormonal induced vaginal atrophy [41]. The review by Rahn et al. [42] proved the superiority of vaginal estrogen over a placebo. The evidence shows reduced vaginal dryness and dyspareunia. Similarly, a study conducted by Weber et al. [43] revealed improvement in genitourinary symptoms such as urinary frequency and urgency and vaginal atrophy in postmenopausal women using topical estrogen. In addition, another study showed an increased epithelial maturation value and improved vaginal health index (VHI), vaginal pH and female sexual functioning index (FSFI) [44]. Moreover, a clinical study conducted by Donders et al. [45] revealed superiority of ultra-low dose estriol and *Lactobacillus acidophilus* vaginal tablets over control group. These postmenopausal breast cancer patients observed significant relief of dyspareunia, soreness and dryness of vaginal mucosa. Interestingly, it has been consistently noted that bacterial vaginitis (BV), vaginal pH, lactobacillary grade (LBG) and vaginal maturation index (VMI) normalize after intervention. Finally, Wills et al. [46] reported in their case-controlled study that intravaginal estradiols elevated circulating E2 levels in postmenopausal women with estrogen receptor-positive breast cancer. This temporarily increase markedly improved symptoms of atrophic vaginitis in breast cancer survivors.

### 4.2. Biological Properties of Vaginal Suppositories

There might be a plausible medical explanation for the efficacy of the following vaginal suppositories. Vitamins A and E are fat-soluble vitamins. All trans-retinoic acids are potent and crucial mediators of numerous organic processes in the human body [47]. Vitamin A plays a crucial role in adequate cellular differentiation, keratinocyte proliferation, tissue growth, reproduction and wound healing [47,48]. Additionally, retinoids enhance the production of fibronectin, collagen type I and extracellular matrix components [49]. It is worth mentioning that all-trans retinoic acids increase the activity of aquaporin 3 gene in vaginal mucosal epithelial cells [47]. These molecules have been proven to increase vaginal lubrication by enhancing water transport across the biological membranes [50]. Moreover, vitamin A is responsible for the accumulation of hyaluronic acid in the upper epithelial layers and stimulation of keratinocyte synthesis [51]. On the contrary, vitamin E exerts antioxidative properties through impeding the formation of reactive oxygen species [52]. Thus, it encourages the repair of the membrane and hence protects cells against damage. Tocotrienols exert an anti-inflammatory effect by downregulating NF-kappaB [53]. Consequently, upregulation of this transcription factor has been implicated in development of chronic tissue inflammation and cancer [54]. Furthermore, tocopherols enhance regulatory activity on potent regulatory and phagocytic mediators such as macrophages and natural killer cells [55]. Hyaluronic acid is a naturally occurring polysaccharide which is responsible for the grouping of proteoglycans in the extracellular matrix of the skin. Furthermore, the biological properties of hyaluronic acid include the support of viscosity and the elasticity of liquid connective tissues as well as the adequate hydration control [56]. Moreover, research attention has gradually increased towards effects of dienestrol on vaginal epithelium. Local estrogen therapy has been shown to alleviate skin dryness, by enhancing sebum secretion. This mechanism is mediated by upregulating the activity of insulin-like growth factor receptors as well as increasing the generation of insulin-like growth factors from fibroblasts [57]. Finally, the synergistic effect of estrogen on the amount of hyaluronic acid and mucopolysaccharides concentration in the skin dermis has been highlighted [58]. This supporting role aids in optimalization of the corneocytes as well as in adequate barrier function [59]. Therefore, it is perhaps unsurprising that the listed vaginal suppositories exert a beneficial effect on healing and management of atrophic vaginitis.

As reported in the available literature, laser therapy might be a promising intervention in management of vaginal side effects after radiotherapy for cancer of endometrium or uterine cervix. Intravaginal laser therapy is a novel non-pharmacological technique, which alleviates symptoms associated with menopausal genital atrophy [60]. Perrone et al. [61] reported a progressive increase in vaginal length following intravaginal non-ablative CO_2_ laser comparing to baseline. Furthermore, to reduce the radiation related vaginal stenosis, vaginal dilators might be recommended. Unfortunately, the available data show poor adherence to treatment, which might be associated with minimal improvement in vaginal symptoms [62,63,64]. Nevertheless, these aforementioned interventions are an interesting approach in management of vulvovaginal symptoms following brachytherapy or radiotherapy.

### 4.3. Strengths and Limitations

It is acknowledged that this study has some limitations. All of the publications in this systematic review were written in English, which might have contributed to the loss of articles published in other languages. Furthermore, our database searches presented heterogeneity in regard to the duration and amount of the vaginal suppositories involved in the intervention. Only four studies met inclusion criteria and were further analyzed. Most of the studies were carried out with an Italian population and only one was from the United States, where the observed response could be related to the genetics of these populations. Moreover, the study conducted by Pitkin et al. [19] is an old study and is the only trial that investigated the effectiveness of estrogen prophylaxis. The study by Galuppi et al. [18] is not a RCT and does not include placebo for the control group. Although, the paper by Dinicola et al. [20] is a RCT, it was conducted in a small population, where the sample size was not calculated and the study was not placebo-controlled. Similarly, the study by Delia et al. [17] is a RCT but not placebo-controlled. Nonetheless, to the best of our knowledge, this is the first review to provide comprehensive information on the vaginal interventions in the context of the vulvovaginal symptoms in patients following radiotherapy or brachytherapy due to endometrial or cervical cancers. Our findings highlight both the importance and the effectiveness of vaginal interventions in alleviating these long-term vulvovaginal side effects. Moreover, we provide a comprehensive presentation of clinically significant key results of various vaginal suppositories on women’s health. The application of lubricants which are rich in hyaluronic acid markedly reduces cytological vaginal atrophy by increasing viscosity and elasticity.

## 5. Conclusions

The diagnosis of gynecological malignancy and the associated treatment certainly has a profound impact on the bodily and emotional aspect of females. The available data about effective management of vulvovaginal late-onset side effects are considerably scarce and usually only describe patient follow-up in the short term.

The application of hyaluronic acid, vitamin A, vitamin E or dienestrol can be considered a positive supplement to gynecological cancer survivors. Thus, a significant improvement in sexual function, vaginal health and self-pleasure can be observed. The aforementioned suppositories reduce the symptoms of long-term vulvovaginal lesions such as dyspareunia, mucosal inflammation, dryness and fibrosis. Moreover, research has established a positive correlation between the vaginal suppositories and decreased vaginal bleeding, fibrosis and cellular atypia. As such, it appears that enhancing the cancer survivors’ quality of life as well as their sexual activity provides them with a heightened sense of identity along with enhanced femininity, value, self-perception and purpose. Unfortunately, the data available in the literature are not sufficient, and large, randomized, placebo-controlled trials are required to prove their effect before introducing these treatments as a standard therapy.

## Figures and Tables

**Figure 1 ijerph-18-03932-f001:**
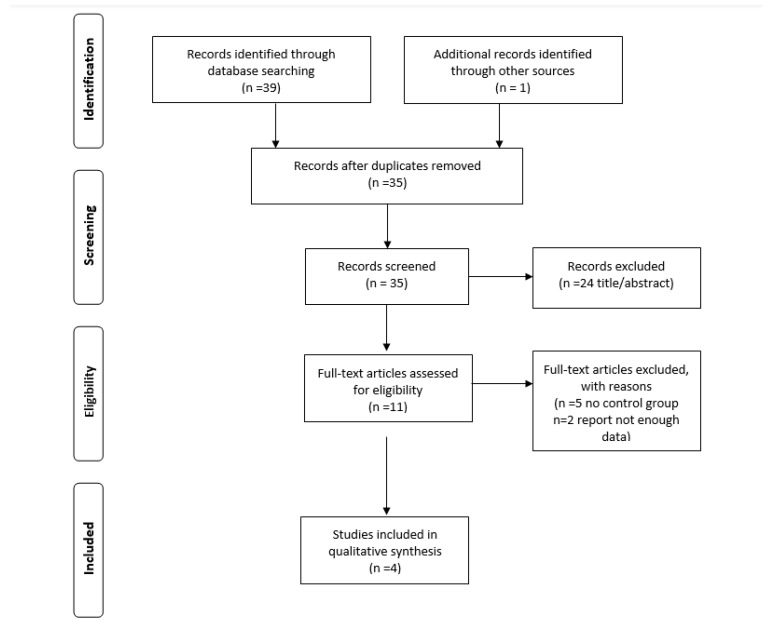
Flow chart of the databases searches on the influence of vaginal interventions on vaginitis and vaginal atrophy after brachytherapy or radiotherapy in patients with malignant endometrial or cervical tumors.

**Figure 2 ijerph-18-03932-f002:**
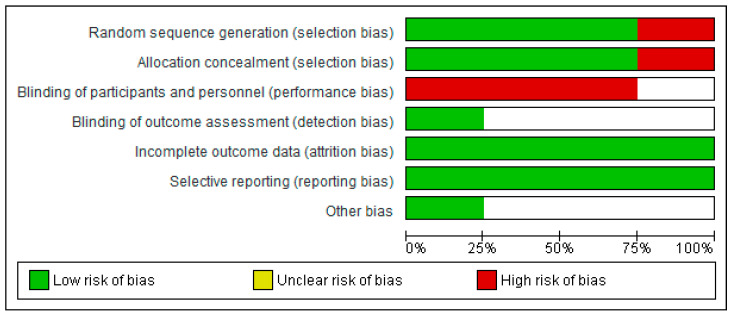
Risk of bias graph: Review authors’ judgements about each risk of bias item presented as percentages across all included studies.

**Table 1 ijerph-18-03932-t001:** Characteristics of enrolled population (*n* = 376).

Source, Year	Country	Trial Type	Sample Size (*n*)		Age (Years) Mean ± SD		Type of Cancer	Staging *	
			Control Group	Intervention Group	Control Group	Intervention Group		Control Group	Intervention Group
Delia et al. 2018 [17]	Italy	RCT	89	88	50.3 ± 10.3	49.7 ± 9.2	Cervical cancer	CC IB–IIA	CC IB–IIA
Dinicola et al. 2015 [20]	Italy	PRS	23	22	38 ± 6	38 ± 6	Cervical cancer	CC IIB–IIIB	CC IIB–IIIB
Galuppi et al. 2011 [18]	Italy	RCT	29	33	66 ± 13.6	67 ± 12.3	Endometrial cancerCervical cancer	EC IC 11	EC IC 9
								EC IIB 3	EC IIB 4
								EC III 1	EC III 1
								CC IB 18	CC IB 15
Pitkin et al. 1971 [19]	USA	RCT	49	43	49.87 ± 13.73	50.04 ± 13.66	Cervical cancer	CC I 9	CC I 10
								CC II 14	CC II 12
								CC III 26	CC III 21
									CC IV 1

RCT, randomized controlled trial; PRS, Prospective randomized Study; EC, Endometrial cancer; CC, cervical cancer; * classified according to the International Federation of Gynecology and Obstetrics (Fédération Internationale de Gynécologie et d’Obstétrique; FIGO).

**Table 2 ijerph-18-03932-t002:** Intervention methods—outcomes of clinical studies.

Study Included	Intervention Time (weeks)	Intervention Dose (mg)	Used Intervention	Method of Evaluation	Key Results
Delia et al. 2018 [17]	5	5	HA	gynecological inspection, Adverse Event Reporting Questions, VAS	↓ vaginal dryness ↓ dyspareunia ↓ mucosal inflammation
		1	Vitamin E		
		1	Vitamin A		
Dinicola et al. 2015 [20]	16	5	HA	biopsy of the vaginal vault, VAS, anamnestic interview, medical examination	↓ dyspareunia ↓ mucosal inflammation ↓ bleeding ↓ fibrosis ↓ cellular atypia
		1	Vitamin E		
		1	Vitamin A		
Galuppi et al. 2011 [18]	14–17	500	alpha-tocopherol acetate	punch biopsy, RTOGSS, VAS, microscopical assessment using histological criteria	↓ mucosal inflammation ↑ vaginal acanthosis
Pitkin et al. 1971 [19]	20–40	7.8	dienestrol	gynecological inspection, historical information	↓ dyspareunia ↓ vaginal caliber ↓ bleeding

VAS, visual analog scale; RTOGSS, Radiation Therapy Oncology Groups Scoring System.

**Table 3 ijerph-18-03932-t003:** Effect of intervention on vaginal parameters.

Source, Year	Dyspareunia		*p*-Value	Dryness		*p*-Value	Inflammation		*p*-Value
	Intervention after	Control after		Intervention after	Control after		Intervention after	Control after	
Delia et al. 2018 [17]	2.46 ± 0.50	0.74 ± 0.43	*p* < 0.001	0.63 ± 0.33	2.43 ± 0.50	*p* < 0.001	0.77 ± 0.25	2.47 ± 0.50	*p* < 0.001
Dinicola et al. 2015 [20]	23%	69%	* p * < 0.05	n/a	n/a	n/a	23%	75%	* p * < 0.05
Galuppi et al. 2011 [18]	1.33 ± 1.55	2.07 ± 1.77	*p* = 0.09	1.17 ± 0.80	0.62 ± 0.62	* p * > 0.05	1.64 ± 0.65	2.21 ± 0.80 *	*p* < 0.05
Pitkin et al. 1971 [19]	23.1%	44.4%	* p * < 0.05	n/a	n/a	n/a	n/a	n/a	n/a

* *p* < 0.05.

**Table 4 ijerph-18-03932-t004:** Effect of intervention on vaginal parameters.

Source, Year	Bleeding		*p*-Value	Fibrosis		*p*-Value
	Intervention after	Control after		Intervention after	Control after	
Delia et al. 2018 [17]	n/a	n/a	n/a	n/a	n/a	n/a
Dinicola et al. 2015 [20]	9%	43%	* p * < 0.05	18%	56%	* p * < 0.05
Galuppi et al. 2011 [18]	n/a	n/a	n/a	2.00 ± 1.15	2.50 ± 0.52	* p * > 0.05
Pitkin et al. 1971 [19]	79.6%	27%	* p * > 0.05	n/a	n/a	n/a

## Data Availability

To get an access to secondary data please contact correspondence author.

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
