# Peer review of "Interventions Preventing Vaginitis, Vaginal Atrophy after Brachytherapy or Radiotherapy Due to Malignant Tumors of the Female Reproductive Organs—A Systematic Review"

_ijerph, 2021, doi:10.3390/ijerph18083932_

Round 1

Reviewer 1 Report

The study approach given by the authors to this manuscript is interesting. One of the advantages is that they evaluate radiotherapy's side effects or brachytherapy in gynecological cancer patients. The studies also used a placebo as a control for treatment with the intervention products.

The disadvantage of the manuscript is that most of the studies were carried out with an Italian population and only one from the United States, where the observed response could be related to the genetics of these populations. In addition to the fact that the United States study is from 1971. Besides, the time of the interventions was different, from 5 weeks to 40 in one study. And in the studies, the same parameters are not evaluated.

MINOR REMARKS

Abbreviations are used in different parts of the manuscript without describing the meaning. I suggest that in each section, the meaning of these abbreviations should be clarified.

In the discussion section in a sentence the authors repeat reference 22 (line 208).

Author Response

We would like to thank the reviewer for her/his thoughtful comments and efforts towards improving our manuscript. The point-to-point response was included below:

  1. Abbreviations are used in different parts of the manuscript without describing the meaning. I suggest that in each section, the meaning of these abbreviations should be clarified.

Thank you for this comment, we corrected accordingly.

  1. In the discussion section in a sentence the authors repeat reference 22 (line 208).

 Thank you for this comment, we corrected accordingly.

Reviewer 2 Report

Radiotherapy vaginitis is a hot spot in gynecological malignancies, overlooked by most clinicians who lack advice and preventative measures for their patients. The problem is hidden by patients who do not speak to their caregivers and ignored by clinicians who do not ask targeted questions to patients. In the literature there is a lack of information on the subject and this review can partially hide this hole. However, to make it more attractive and more useful for the clinician who will consult it, some tips may be useful.

- the date of the revision start is missing (page 2 line 92)

- The vaginal laser represents a new approach as a therapy should be included in the analysis, I believe there is only one article and one trial published

“Perrone AM, Tesei M, Ferioli M, De Terlizzi F, Della Gatta AN, Boussedra S, Dondi G, Galuppi A, Morganti AG, De Iaco P. Results of a Phase I-II Study on Laser Therapy for Vaginal Side Effects after Radiotherapy for Cancer of Uterine Cervix or Endometrium. Cancers (Basel). 2020 Jun 21;12(6):1639. doi: 10.3390/cancers12061639. PMID: 32575821; PMCID: PMC7352893”.

“Athanasiou S, Pitsouni E, Grigoriadis T, Michailidis G, Tsiveleka A, Rodolakis A, Loutradis D. A study protocol of vaginal laser therapy in gynecological cancer survivors. Climacteric. 2020 Feb;23(1):53-58. doi: 10.1080/13697137.2019.1646720. Epub 2019 Sep 2. PMID: 31474161.”

- To make the systematic review more interesting, adding dilator studies considering they are not many shouldn't take much time. I did a short research, there are 45 studies in total, but the useful ones will be few ….

- “disagreements regarding study selection were cross-checked by a third reviewer” could also be “Two authors independently applied the risk of bias tools and resolved differences by discussion” (page 3, line 126).

- Visual analogue scale is generally defined as VAS not VSA

- Summarizing many parts of the discussion is too long and repetitive in some places, and adding the new talking points will also make it possible to lengthen the part of the results that appears too short.

Author Response

We would like to thank the reviewer for her/his thoughtful comments and efforts towards improving our manuscript. The point-to-point response was included below:

  1. the date of the revision start is missing (page 2 line 92)

Thank you for this comment, we conducted the systematic search between October and November 2020 as stated in the text

  1. The vaginal laser represents a new approach as a therapy should be included in the analysis, I believe there is only one article and one trial published

“Perrone AM, Tesei M, Ferioli M, De Terlizzi F, Della Gatta AN, Boussedra S, Dondi G, Galuppi A, Morganti AG, De Iaco P. Results of a Phase I-II Study on Laser Therapy for Vaginal Side Effects after Radiotherapy for Cancer of Uterine Cervix or Endometrium. Cancers (Basel). 2020 Jun 21;12(6):1639. doi: 10.3390/cancers12061639. PMID: 32575821; PMCID: PMC7352893”.

“Athanasiou S, Pitsouni E, Grigoriadis T, Michailidis G, Tsiveleka A, Rodolakis A, Loutradis D. A study protocol of vaginal laser therapy in gynecological cancer survivors. Climacteric. 2020 Feb;23(1):53-58. doi: 10.1080/13697137.2019.1646720. Epub 2019 Sep 2. PMID: 31474161.”

 Thank you for this valuable comment and recommendation of very interesting publication by Perrone et al. entitled “Results of a Phase I-II Study on Laser Therapy for Vaginal Side Effects after Radiotherapy for Cancer of Uterine Cervix or Endometrium”. In this study authors determined the effect of laser therapy on vaginal length in women undergoing pelvic radiotherapy. Unfortunately, this publication does not contain crucial for our systematic review information regarding changes in dyspareunia, dryness, inflammation, bleeding or fibrosis following intervention. That is why we did not include this study to our systematic review. However, we included both articles in the discussion section.

  1. To make the systematic review more interesting, adding dilator studies considering they are not many shouldn't take much time. I did a short research, there are 45 studies in total, but the useful ones will be few ….

Thank you for this comment. We conducted database search and found only 3 articles which might have been relevant to our systematic review. After careful analysis, two of these studies measured the adherence of vaginal dilator use (where one was a pilot study), and one study focused on dimensions of vaginal canal following dilator use and unfortunately none of these studies focused on crucial for our systematic review information regarding changes in dyspareunia, dryness, inflammation, bleeding or fibrosis following intervention. That is why we did not include this study to our systematic review. However, we included these articles in the discussion section.

  1. “disagreements regarding study selection were cross-checked by a third reviewer” could also be “Two authors independently applied the risk of bias tools and resolved differences by discussion” (page 3, line 126).

Thank you for this suggestion, we corrected accordingly.

  1. Visual analogue scale is generally defined as VAS not VSA

Thank you for this comment, we corrected accordingly.

  1. Summarizing many parts of the discussion is too long and repetitive in some places, and adding the new talking points will also make it possible to lengthen the part of the results that appears too short.

Thank you for this comment. We shortened the discussion section and removed repetitive parts. Moreover, we added a paragraph about laser therapy as well as vaginal dilators as an interesting approach in management of vulvovaginal symptoms following brachytherapy or radiotherapy.

Reviewer 3 Report

The Authors performed a systematic review (SR) to evaluate the effectiveness of interventions preventing vaginitis, vaginal atrophy after radiotherapy in patients affected by gynecologic cancers. The topic is very important and relevant for clinical practice. However, I have many criticisms about the manuscript.

Major comments

  1. The SR was not registered in the PROSPERO website.
  2. The main outcomes and measures should be reported in the Materials and Methods section.
  3. The quality assessment of the studies included in the SR should be performed. The Cochrane Collaboration Risk of Bias Tool could be used for the objective assessment of the RCTs.
  4. There is a large heterogeneity between the four studies of the SR. Furthermore, all the studies have severe limitations:

- The study by Pitkin et al., 1971 is very old and is the only trial that investigated the effectiveness of estrogen prophylaxis.

- The study bt Galuppi et al., 2011 is not a randomized controlled trial and does not include placebo for the control group.

- The study bt Dinicola et al., 2015 is a randomized trial but the sample size was not calculated and is not placebo-controlled. Moreover, the study was not registered.

- The study by Delia et al., 2015 is a randomized trial but not placebo-controlled. Also, this study was not registered.

  1. The statistical methods were not described.

Minor comments

Line 172

The Authors state: “All of the studies showed the superiority of vaginal interventions products when  compared with a placebo.”

Unfortunately, the studies were not placebo-controlled, so that they showed the superiority of the vaginal products versus no treatment.

 Line 189

The Discussion is too long, and it should focus on the results of the SR.

Line 314

The SR has many limitations and should be completely reported.

Line 327

The conclusions should be more concise and report a clear message from the main results of the SR. 

Line 358.

The results of the SR do not allow for any firm conclusions due to the flaws in the selected studies. All the studies concluded that large trials are necessary to confirm the effectiveness of the vaginal suppositories in preventing the side-effects of radiotherapy.

Therefore, the right conclusion of this review should be that the data available in the literature is not sufficient to recommend these treatments, and that large randomized, placebo-controlled trials are required before introducing them into clinical practice.

Author Response

We would like to thank the reviewer for her/his thoughtful comments and efforts towards improving our manuscript. The point-to-point response was included below:

Major comments

  1. The SR was not registered in the PROSPERO website.

  1. Thank you for this comment, the PROSPERO is known for us and we have used it for different reviews to get first overview on selected topic and methodology. This review is part of PhD thesis and according the rules at the University any part cannot be published before is officially finished. Nevertheless, the working document - “protocol” was prepared at the beginning and consulted in broadened group of professionals from the field and we followed accordingly to PROSPERO rules.

  1. The main outcomes and measures should be reported in the Materials and Methods section.

Thank you for this comment. It was added in materials and methods.

  1. The quality assessment of the studies included in the SR should be performed. The Cochrane Collaboration Risk of Bias Tool could be used for the objective assessment of the RCTs.

It was added at the Fig 2

  1. There is a large heterogeneity between the four studies of the SR. Furthermore, all the studies have severe limitations:

- The study by Pitkin et al., 1971 is very old and is the only trial that investigated the effectiveness of estrogen prophylaxis.

- The study bt Galuppi et al., 2011 is not a randomized controlled trial and does not include placebo for the control group.

- The study bt Dinicola et al., 2015 is a randomized trial but the sample size was not calculated and is not placebo-controlled. Moreover, the study was not registered.

- The study by Delia et al., 2015 is a randomized trial but not placebo-controlled. Also, this study was not registered.

 Thank you for this comment, we included these limitations in our study. It is difficult to contact each of the authors to get more information than these included in the paper.

  1. The statistical methods were not described.

Due to heterogeneity of the selected studies SLR was not finalized by metanalysis that’s why we omitted statistical analysis section. 

Minor comments

  1. Line 172 The Authors state: “All of the studies showed the superiority of vaginal interventions products when  compared with a placebo.” Unfortunately, the studies were not placebo-controlled, so that they showed the superiority of the vaginal products versus no treatment.

 Thank you for this comment, we corrected accordingly.

  1.  Line 189 The Discussion is too long, and it should focus on the results of the SR. Thank you for this comment. We shortened the discussion section and removed repetitive parts focusing on the results of systematic review.

8.Line 314 The SR has many limitations and should be completely reported.

 Thank you for this comment, we reported limitations of the study.

9.Line 327 The conclusions should be more concise and report a clear message from the main results of the SR. 

Thank you for this comment, we shortened the conclusion section and focused on the main results of the systematic review.

  1. Line 358. The results of the SR do not allow for any firm conclusions due to the flaws in the selected studies. All the studies concluded that large trials are necessary to confirm the effectiveness of the vaginal suppositories in preventing the side-effects of radiotherapy. Therefore, the right conclusion of this review should be that the data available in the literature is not sufficient to recommend these treatments, and that large randomized, placebo-controlled trials are required before introducing them into clinical practice.

Thank you for this comment, we included a sentence, where large, randomized, placebo-controlled trials are required before introducing these treatments into clinical practice.

Round 2

Reviewer 2 Report

Weel done! Acceped

Author Response

Thank you very much for your positive evaluation!

Reviewer 3 Report

The authors have satisfactorily addressed most of my comments and the manuscript has been significantly improved. 
The only remaining concerns I have are with statistical methods. The methods used for the statistical analysis of the Table 3 should be described in the “Statistical analysis” section.

Author Response

We would like to thank the reviewer for her/his thoughtful comments and efforts towards improving our manuscript. The point-to-point response was included below:

The authors have satisfactorily addressed most of my comments and the manuscript has been significantly improved. 

The only remaining concerns I have are with statistical methods. The methods used for the statistical analysis of the Table 3 should be described in the “Statistical analysis” section.

Thank you for this comment. The suggested change was done.